# Maternal dietary patterns, breastfeeding duration, and their association with child cognitive function and head circumference growth: A prospective mother–child cohort study

David Horner[1]*, Jens Richardt Møllegaard Jepsen[2,3], Bo Chawes[1], Rebecca Vinding[1], Julie B. Rosenberg[1,2,4], Parisa Mohammadzadeh[1,2,4], Yang Luo[1], Birgitte Fagerlund[2,5], Trine Flensborg-Madsen[6], Thomas Ragnar Wood[7], Janine F. Felix[8,9], Louise Monnerup[1], Birte Y. Glenthøj[2,4], Klaus Bønnelykke[1], Bjørn H. Ebdrup[2,4☯], Jakob Stokholm[1,10☯], Morten Arendt Rasmussen[1,10☯]

1 COPSAC, Copenhagen Prospective Studies on Asthma in Childhood, Herlev and Gentofte Hospital, University of Copenhagen, Gentofte, Denmark, 2 Center for Neuropsychiatric Schizophrenia Research (CNSR) & Centre for Clinical Intervention and Neuropsychiatric Schizophrenia Research (CINS), Mental Health Centre Glostrup, University of Copenhagen, Glostrup, Denmark, 3 Mental Health Centre for Child and Adolescent Psychiatry - Research unit, Mental Health Services in the Capital Region of Denmark, Copenhagen, Denmark, 4 Department of Clinical Medicine, Faculty of Health and Medical Sciences, University of Copenhagen, Copenhagen, Denmark, 5 Department of Psychology, University of Copenhagen, Copenhagen, Denmark, 6 Unit of Medical Psychology, Department of Public Health, University of Copenhagen, Copenhagen, Denmark, 7 Department of Pediatrics, Division of Neonatology, University of Washington School of Medicine, Seattle, Washington, United States of America, 8 The Generation R Study Group, Erasmus MC, University Medical Center Rotterdam, Rotterdam, the Netherlands, 9 Department of Paediatrics, Erasmus MC, University Medical Center Rotterdam, Rotterdam, the Netherlands, 10 Section of Food, Microbiology and Fermentation, Department of Food Science, University of Copenhagen, Copenhagen, Denmark

☯ These authors contributed equally to this work.
* david.horner@dbac.dk

## Abstract

### Background

Early life is a critical period for neurodevelopment, where factors such as maternal nutrition and breastfeeding duration significantly impact the growth of head circumference and cognitive development in children. Our study aimed to explore the associations between maternal dietary patterns during pregnancy, duration of breastfeeding, and their impacts on child head circumference and cognitive outcomes.

### Methods and findings

Our study utilised data from the Copenhagen Prospective Studies on Asthma in Childhood 2010 cohort, which enrolled 700 mother–child pairs between 2008 and 2010 with 86% clinical follow-up at age 10. Pregnancy dietary patterns, described as 'Varied' and 'Western,' were derived from food frequency questionnaires and used to model quantitative metabolite scores via sparse partial least squares modelling of blood metabolome data.

**Data availability statement:** Participant-level personally identifiable data are protected under the Danish Data Protection Act and European Regulation 2016/679 of the European Parliament and of the Council (GDPR), which prohibit distribution even in pseudo-anonymized form. Participant-level data can be made available under a data transfer agreement as part of a collaboration effort. Data requests should be directed to COPSAC's Data Protection Officer (DPO), Ulrik Ralfkiaer, PhD, at administration@dbac.dk, for researchers who meet the criteria for access to confidential data. To ensure compliance with these regulations, all data transfer agreements will adhere to institutional and ethical guidelines.

**Funding:** All funding received by COPSAC is listed on www.copsac.com. This work was supported by the Lundbeck Foundation (Grant no. R16-A1694 to KB and R269-2017-5 to JRMJ, RV, JR, and PM); the Ministry of Health (Grant no 903516 to KB); the Danish Council for Strategic Research (Grant no 0603-00280B to KB); the Capital Region Research Foundation (to BC, KB, and JS); and the European Research Council (ERC) under the European Union's Horizon 2020 research and innovation programme (grant agreement No. 946228 to BC). MAR is funded by the Novo Nordisk Foundation (Grant no NNF21OC0068517). The funders had no role in study design, data collection and analysis, decision to publish, or preparation of the manuscript.

**Competing interests:** Authors have read the journal's policy, and have the following competing interests: BE is a member of the Advisory Board of Eli Lilly Denmark A/S, Janssen-Cilag, Lundbeck Pharma A/S, and Takeda Pharmaceutical Company Ltd. BE has also received lecture fees from Bristol-Myers Squibb, Boehringer Ingelheim, Otsuka Pharma Scandinavia AB, Eli Lilly Company, and Lundbeck Pharma A/S. The remaining authors have declared that no competing interests exist. The funding agencies had no role in the design or conduct of the study; the collection, management, or interpretation of the data; or the preparation, review, or approval of the manuscript. No pharmaceutical company was involved in the study.

**Abbreviations:** ACME, Average Causal Mediation Effect; COPSAC2010, COpenhagen Prospective Studies on Asthma in Childhood

Cognitive development was assessed using the Bayley Scales of Infant Development at 2.5 years and the Wechsler Intelligence Scale for Children at age 10. Head circumference was measured from 20 weeks gestation to 10 years, and calibrated using related anthropometric measures. Growth trajectories were evaluated using linear mixed models and latent class trajectory models. Parental and child genetic influences for cognition and head circumference were controlled by including polygenic risk scores derived from genomic data. We found that a Western dietary pattern during pregnancy was associated with lower cognitive scores at age 2.5 ($\beta$ –1.24 [–2.16, –0.32], $p = 0.008$) and reduced head circumference growth ($p$-interaction < 0.0001). We found that a Varied dietary pattern during pregnancy was associated with higher estimated intelligence quotient (IQ) at age 10 ($\beta$ 1.29 [0.27, 2.3], $p = 0.014$). Additionally, head circumference growth was associated with higher cognitive scores at age 10 ($\beta$ 3.40 [1.21, 5.60], $p = 0.002$), and it partly mediates the association between the Varied dietary pattern and estimated IQ (proportion mediated 13.5% [0.01, 0.71], $p = 0.034$). Extended breastfeeding duration was also independently associated with increased head circumference growth ($p$-interaction < 0.0001). These patterns and correlations were consistent even after adjusting for potential confounders and accounting for genetic influences.

## Conclusions

Our findings reveal that a Western dietary pattern during pregnancy is associated with lower cognitive scores at age 2.5 and decreased head circumference growth, suggesting potential adverse impacts on early neurodevelopment. Conversely, a Varied dietary pattern is linked with a higher estimated IQ at age 10, with head circumference growth contributing to this positive outcome. These findings highlight the critical role of maternal nutrition during pregnancy, and duration of breastfeeding, in promoting optimal neurodevelopmental outcomes. Effective public health strategies should therefore focus on enhancing maternal dietary practices to support better cognitive and physical development in children.

## Author summary

### Why was this study done?

- Early neurodevelopment is influenced by maternal nutrition during pregnancy and breastfeeding, but the specific impacts of dietary patterns on cognitive outcomes and head circumference growth, an indicator of early brain development, in children remain unclear.

- The study sought to investigate the role of maternal diet and breastfeeding in shaping cognitive outcomes and head circumference growth in children.

### What did the researchers do and find?

- The study analysed 700 mother–child pairs from the COpenhagen Prospective Studies on Asthma in Childhood 2010 (COPSAC2010) cohort, assessing maternal dietary patterns during pregnancy and breastfeeding duration.

2010; CCS, Cognitive composite score; COPSYCH, The COpenhagen Prospective Study on Neuro-PSYCHiatric Development study; Bayley-III, Danish Bayley Scales of Infant and Toddler Development; FFQ, Food frequency questionnaire; FSIQ, Full Scale Intelligence Quotient; GWAS, Genome-wide association study; IQ, Intelligence quotient; PCA, Principal component analysis; PRS, Polygenic risk score; sPLS, Sparse partial least square; WISC-IV, Wechsler Intelligence Scale for Children-fourth edition.

- In pregnancy, a Western dietary pattern was linked to reduced early cognitive scores and head circumference growth, while a Varied dietary pattern and longer breastfeeding were associated with improved head circumference growth and IQ.

- Head circumference growth partially mediated the positive impact of a Varied dietary pattern on cognitive outcomes.

## What do these findings mean?

- These findings highlight the importance of a balanced maternal diet during pregnancy and breastfeeding to optimise child neurodevelopment.

- The study's observational design means residual confounding cannot be entirely excluded, and results are most applicable to developed countries.

## Introduction

Cognition is a complex trait and encompasses the ability to learn, reason, and solve problems [1]. Childhood intelligence is an important factor in predicting future outcomes such as health [2], education [3], and occupation [4]. While heritability estimates for intelligence are high in adult populations [5], estimates are lower in childhood, increasing through adolescence [6]. This suggests that environmental factors may exert a greater influence in early life [7,8].

Head circumference is a reliable indicator of brain growth during early life and is strongly correlated with brain volume [9]. The perinatal period is a crucial time for brain development and approximately 90% of adult brain volume is reached by age 5 [10], suggesting that this is a sensitive period of neurodevelopment. Accordingly, head circumference has been associated with cognitive abilities in both childhood [11] and adulthood [12,13]. Longitudinal measurements of head circumference growth are strongly associated with cognition in children born preterm [14,15]; however, studies are limited in the general population [16].

Given this critical period of rapid brain expansion during pregnancy [17,18], the interplay between nutrition, genetics, and environmental factors is essential for supporting healthy neurodevelopment [19]. Although genetics play a significant role in shaping neurodevelopmental outcomes, including cognitive abilities and head circumference [20,21], early-life environmental influences, such as maternal and child nutrition, may be influential during early development. Maternal diet, as assessed through isolated nutrients or food groups, has thus been shown to have long-term effects on cognitive development, intelligence, and behavioural outcomes [22,23]. Focussing on overall dietary patterns rather than single nutrients, studies have found that children of mothers following unhealthy dietary clusters during pregnancy have a lower intelligence quotient (IQ) at age 8 compared to those born to mothers adhering to healthier, fruit- and vegetable-rich diets [46]. Nutrition in early life, including breastfeeding, is another contributing factor to child neurodevelopment [24,25]. However, previous research has not adequately disentangled the potential independent, additive, or pleiotropic effects of maternal and child dietary intake on brain morphological or cognitive outcomes. Furthermore, whilst assessing individual nutrients or food groups is informative [26], this approach overlooks the potential synergistic effects of nutrients [27]. Dietary patterns, however, offer a broader perspective on nutrient intake, capturing overall dietary habits and their potential interactions. Yet, reliance on food frequency questionnaires (FFQs) introduces recall bias [28], and few studies have mitigated this limitation by incorporating objective biomarkers [29].

In this study, we leverage data from the COpenhagen Prospective Studies on Asthma in Childhood 2010 (COPSAC2010) cohort of 700 mother–child pairs. We hypothesise that pregnancy and early-life dietary influences are associated with cognitive development and head circumference growth in childhood. Our dietary exposures include objective measures of pregnancy dietary patterns via blood metabolome modelling, and duration of breastfeeding. We utilise comprehensive neuropsychological assessments to evaluate cognition, and an extensive longitudinal dataset of up to 15 head circumference measurements, up to the age of 10. Furthermore, we examine the contributions of maternal and child genetic influences, and child metabolome dietary scores to explore these interrelationships.

## Methods

### Study population

The COPSAC2010 cohort includes 700 mother–child pairs with extensive phenotyping from 15 clinical visits and exposure assessments beginning in pregnancy [30]. The COpenhagen Prospective Study on Neuro-PSYCHiatric Development study (COPSYCH) was nested within COPSAC2010 at age 10 years [31].

### Dietary exposures

Our main exposures included dietary pattern metabolite scores, assessed from global blood metabolomics profiles taken at 24 weeks gestation (HD4 platform, Metabolon (NC, USA)), derived from sparse partial least squares (sPLS) modelling, and breastfeeding duration (length of breastfeeding in days). Dietary pattern metabolite scores were modelled on dietary patterns during pregnancy, as identified from a validated 360-item semi-quantitative FFQ completed by mothers at recruitment (24 weeks gestation), with a 1-month recall period [32,33]. Principal component analysis (PCA) identified dietary patterns based on calculated estimates of energy, macronutrients, and micronutrients derived from pregnancy FFQs. The first principal component reflected a "Varied dietary pattern" (44.3% of variance), characterised by positive associations with a wide range of FFQ-derived food groups, including whole grains, fish, eggs, and nuts. The second principal component reflected a "Western dietary pattern" (10.7% of variance), characterised by positive associations with FFQ-derived food groups such as animal fats, refined grains, and high-energy drinks, and negative associations with fruits, fish, and vegetables (S1 Fig) [30]. Nutrient intake data and principal component loadings are detailed S1 Table. These dietary patterns were used to derive the metabolite scores in the sPLS models. Breastfeeding duration, measured as the length of breastfeeding in days, was also considered as a dietary exposure. Further details regarding the blood metabolomics analysis, derivation of dietary pattern metabolite scores via sPLS modelling, and data normalisation procedures for breastfeeding duration can be found in the Supplementary Methods. In subanalysis, dietary pattern metabolite scores were modelled for other maternal and child blood metabolome time points (maternal 1-week postpartum and children at 6 months, 18 months, and 6 years), trained on the pregnancy time point. Metabolite scores and breastfeeding length were internally z-scored at each time point, thus reflecting a change of 1 standard deviation (SD) within the population. When assessing the relationships between specific dietary exposures and neurodevelopmental outcomes, exposures were evaluated independently.

### Cognitive ability

Cognition was assessed at 2.5 years [34] as reflected in the cognitive composite score (CCS) derived from the 3rd edition of the Danish [35] Bayley Scales of Infant and Toddler Development (Bayley-III) [36]. CCS was standardised with age-corrected means of 100 and a SD of 15,

ranging from 50 to 150. For cognition at 10 years, the Full Scale Intelligence Quotient (FSIQ) was estimated, based on selected tests from the fourth edition of the Wechsler Intelligence Scale for Children-fourth edition (WISC-IV) [37], including Matrices, Vocabulary, Digit Span, Letter-Number Sequencing, Coding, and Symbol Search. Estimated FSIQ likewise utilised age-corrected means of 100 and a standard deviation of 15, ranging from 40 to 160. Previous studies have reported moderate longitudinal correlations between CCS and FSIQ [38].

## Head circumference growth

Head circumference was measured using a tape measure at each of the 14 scheduled clinical visits, with the largest diameter serving as the measurement parameter (1 week–10 years). Foetal head circumference measurements were added from the week 20 prenatal screening programme using the Danish Føto-database [39].

In our main analysis, calibration of child head circumference was performed to account for potential confounding of related growth metrics. This was achieved by regressing head circumference against age (gestational age-adjusted), sex, length (0–18 months) or height (2–10 years), weight, and waist circumference at each visit. The residuals from this regression model were then used to create a calibrated measure of head circumference for each visit, and subsequently $z$-scored. Foetal head circumference was likewise calibrated for gestational age, sex, femur length, and abdominal circumference at measurement. Covariates with extreme outliers (those exceeding ±6 SDs) and deviations exceeding ±2 SD from an individual's global growth trajectory were removed prior to calibration. Missing calibration covariates were imputed from linear mixed models on individual child growth trajectories. Head circumference measurements ± 3 SD were visually inspected to ensure data was in keeping with repeated growth measurements. In sensitivity analysis, head circumference measurements that lay outside of 2 SD of an individual's global growth trajectory were removed (1.0%) and separately, head circumference was adjusted only for child age and sex.

We modelled head circumference longitudinally using latent class growth trajectories and linear mixed models. We considered latent class models with 2–5 latent classes. The optimal fit was a 3-class linear growth model, selected based on the smallest Akaike information criterion, Bayesian information criterion, and biological interpretation. The "lcmm" R-package was used [40]. Linear mixed models were applied with variable slopes and intercepts, the slopes and intercepts were treated as random effects, allowing for modelling of individual head circumference growth and head circumference in early life, respectively. Accordingly, these random effects were extracted, the "lme4" R-package was used for modelling [41] and both slopes and intercepts were scaled (estimates are interpreted as per SD change).

## Polygenic risk scores

Maternal and child genotypes in the COPSAC2010 cohort were analysed using the Illumina Infinium HumanOmniExpressExome BeadChip. To calculate a polygenic risk score (PRS) for intelligence, a genome-wide association study (GWAS) from the UK Biobank was utilised ($n = 391,124$) measuring fluid intelligence [42]. For child head circumference PRS scores, we utilised a GWAS derived from child head circumference measurements ($n = 29,192$) [21]. As COPSAC2010 participated in this consortium study, we contacted the authors who re-ran the GWAS with COPSAC2010 excluded. The PRS-CS method, which integrates millions of SNPs across the genome for enhanced precision, was applied. This method is well-validated in European populations and ensures that the PRS captures genome-wide genetic predispositions rather than being limited to specific loci. The strong genetic basis of intelligence and head circumference, combined with the robust GWAS methodologies, supports the relevance

of these PRS to our outcomes. For head circumference, parameters were manually tuned during analysis using a grid search approach to optimise the global scaling factor. PRS were *z*-scored to mean 0 and SD 1 for interpretation.

## Information on covariates

Potential confounders were identified with a directed acyclic graph and existing literature [13,43–46]. In multivariable analysis, we included the following covariates: pre-pregnancy maternal body mass index (BMI), child sex, birthweight, gestational age, smoking during pregnancy, antibiotic use during pregnancy, pre-eclampsia during pregnancy, household income at birth (low (<€50,000), medium (€50,000–€110,000), high (>€110,000)), maternal education level at birth (numerically coded as: 1 (elementary and college education), 2 (medium education), 3 (university education)) and maternal age. All missing covariate data were imputed for analysis, based on other available covariate data.

To control for genetic confounding, we included maternal and child intelligence and head circumference genetic scores in further multivariable analysis. Likewise in sensitivity analysis, we further adjusted for any child neurodevelopmental disorders at 10 years, a child Western dietary pattern at 10 years [29], parental head circumference and outcomes from two prenatal randomised controlled trials: n3-long-chain polyunsaturated fatty acids supplementation and high-dose vitamin D during pregnancy [30].

## Analysis plan

This study did not have a prospective analysis plan established prior to its commencement. Hypothesis-driven analyses were planned in mid-2023, based on existing literature and the study's specific objectives. No data-driven changes to the analysis were made, and no modifications were implemented in response to peer review comments.

## Statistical analysis

Linear regression models were used to examine the association between dietary variables and cognitive scores, as well as the association between head circumference and cognitive scores. Linear mixed models were used to assess the associations between dietary variables and head circumference growth, incorporating interaction terms with time and random effects to account for individual trajectories over multiple visits. In subanalysis, we employed nested linear and linear mixed models, comparing them using likelihood ratio tests (using the anova() function in R, specifying a Chi-square test) to assess the statistical significance of model differences. To handle the issue of missing metabolite scores (14.7% missing), essential for balanced anova comparison modelling, we imputed missing data using the missMDA package [47]. Causal mediation analysis was conducted with multivariable models using the mediation package in R (v4.5.0), employing 1,000 iterations. We removed the second twin child in twin pairs ($n = 5$) from our analysis, since the exposures of interest were not independent. The statistical analyses were designed in 2022 based on hypotheses derived from existing literature and the comprehensive data in the COPSAC2010 cohort. All data analyses were performed with the statistical software R version 4.1.1. Other R packages utilised in this analysis include tidyverse (v1.3.1), dplyr (v1.0.10), broom (v0.7.12), psych (v2.1.9), stats (v4.4.1), scatterplot3d (v.3.41), ggpubr (v0.4.0), lubridate (v1.8.0) and tableone (v0.13.0).

## Inclusion and ethics

The study was approved by The Local Ethics Committee (H-B-2008-093), and the Danish Data Protection Agency (2015-41-3696). Written informed consent was obtained from participants at cohort recruitment, prior to any study-related procedures.

## Results

### Baseline characteristics and dietary exposures

Dietary exposures included dietary pattern metabolite scores derived from maternal blood metabolome data (available for 684 mothers) and breastfeeding duration data (available for 687 mother–child pairs). Outcomes assessed were cognition, evaluated in 627 (90.2%) children at 2.5 years and 586 children (84.3%) at 10 years, and head circumference growth, based on 9,501 measurements collected from 20 weeks gestation to 10 years (8.5% data missing). Each child in the main analysis contributed a median of 15 head circumference measurements (interquartile range 13–15). There were no significant differences in baseline characteristics between participants and non-participants at the 10-year visit (S2 Table) nor between male and female children (S3 Table).

A Western dietary pattern metabolite score in pregnancy was characterised with increased intakes of animal fats, refined grains, and high-energy drinks, and a lower score was associated with increased intakes of fish, breakfast cereals, fruits, and vegetables (Fig 1A). A Varied dietary pattern metabolite score was characterised with increased intakes of coffee and breakfast cereals, and lower intakes of snacks and energy drinks (Fig 1B). The mean (SD) breastfeeding duration was 247 days (sd = 165) (Fig 1F).

The Western dietary pattern was associated with numerous covariates (Table 1). Furthermore, a Western dietary pattern was negatively associated with a varied dietary pattern ($p < 0.001$) and breastfeeding duration ($p < 0.001$), reflecting the interconnected nature of these

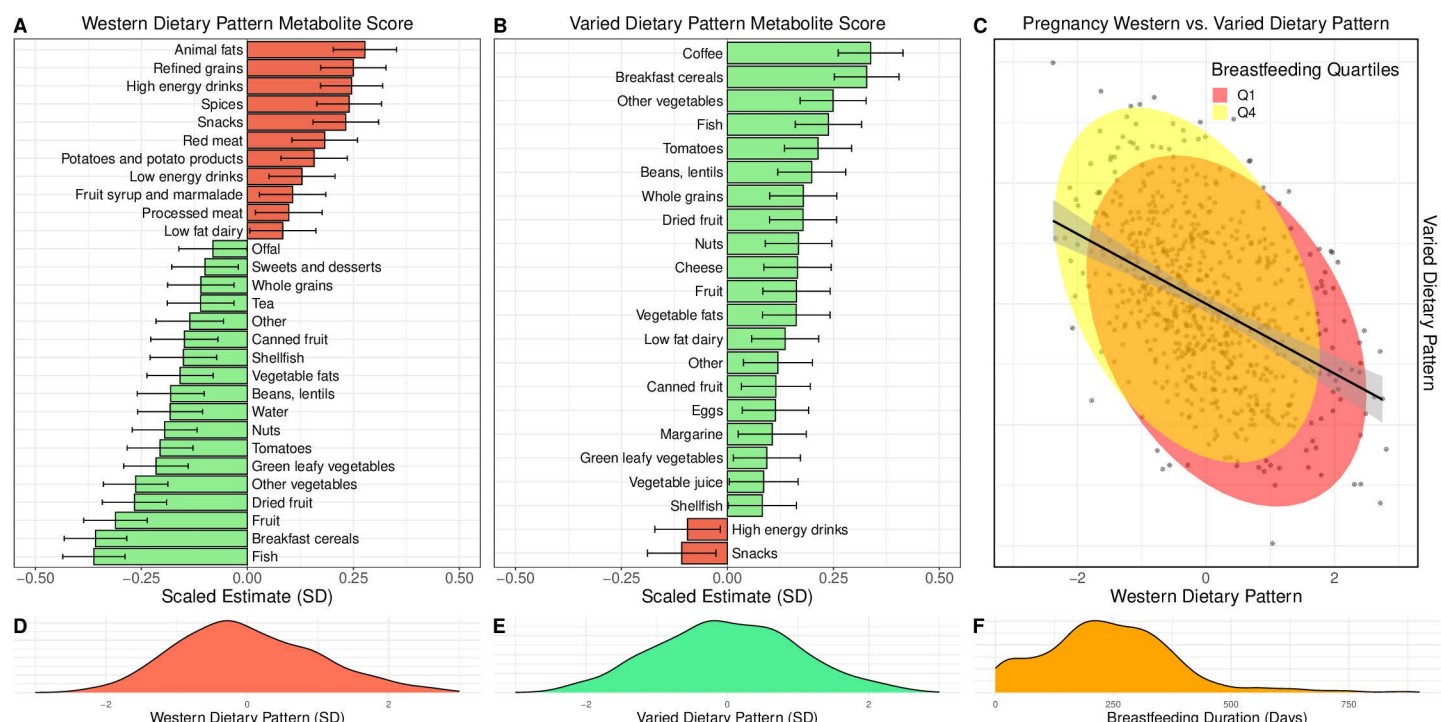

**Fig 1. Associations of the Western and Varied dietary patterns metabolite scores during pregnancy with food frequency derived food groups.** This figure presents the associations between the Western and Varied dietary patterns metabolite scores during pregnancy and food frequency-derived food groups. Panels **A** and **B** depict these associations for the Western and Varied dietary patterns, respectively. Panel **C** presents a scatter plot with a regression line showing the strong association between Western and Varied dietary patterns, with ellipses for high and low breastfeeding duration. Panels **D**, **E**, and **F** display the distribution of our data in density plots for the Western dietary pattern metabolite score in pregnancy, Varied dietary pattern metabolite score in pregnancy, and breastfeeding duration, respectively.

**Table 1. Covariates and dietary exposures stratified by high and low Western dietary pattern metabolite scores during pregnancy.** This table presents the model covariates and dietary exposures, stratified by high and low (median split) Western dietary pattern metabolite scores during pregnancy (*n* = 684). It provides a comprehensive overview of the cohorts' characteristics, including differences in other dietary exposures and covariates stratified by high and low Western dietary pattern groups.

| Cohort characteristics | Low Western Dietary Pattern | High Western Dietary Pattern | p |
|---|---|---|---|
| *n* = | 342 | 342 | |
| Child Sex (%) | 174 (50.9) | 176 (51.5) | 0.939 |
| Maternal pre-pregnancy BMI (mean (SD)) | 23.4 (3.4) | 25.8 (4.96) | <0.001 |
| Smoking during pregnancy (%) | 9 (2.6) | 43 (12.6) | <0.001 |
| Alcohol during pregnancy (%) | 45 (13.2) | 24 (7.0) | 0.011 |
| Gestational age in (days) (mean (SD)) | 279.7 (12.1) | 278.9 (10.5) | 0.378 |
| Birthweight (kg) (mean (SD)) | 3.54 (0.55) | 3.56 (0.53) | 0.568 |
| Antibiotics during pregnancy (%) | 110 (32.3) | 137 (40.1) | 0.041 |
| Pre-eclampsia during pregnancy (%) | 14 (4.1) | 16 (4.7) | 0.858 |
| Maternal Educational Level at Birth (%) | | | <0.001 |
| Low (primary, secondary, or college graduate (%) | 11 (3.2) | 39 (11.4) | |
| Medium (tradesman or bachelor's degree (%) | 191 (55.8) | 243 (71.1) | |
| High (Master's degree) (%) | 140 (40.9) | 60 (17.5) | |
| Income type (%) | | | <0.001 |
| Low (<50,000 euro) | 20 (5.8) | 38 (11.1) | |
| Medium (50,000–110,000 euro) | 155 (45.3) | 213 (62.3) | |
| High (>110,000 euro) | 167 (48.8) | 91 (26.6) | |
| Maternal age (mean (SD)) | 33.0 (4.2) | 31.6 (4.4) | <0.001 |
| Varied Dietary Pattern (mean (SD)) | 0.39 (0.90) | −0.39 (0.95) | <0.001 |
| Breastfeeding duration (mean (SD)) | 272.44 (180) | 222.26 (147) | <0.001 |
| Fish Oil RCT (%) | | | 0.247 |
| Yes | 163 (47.7) | 180 (52.6) | |
| No | 179 (52.3) | 161 (47.1) | |
| Not Participated | 0 (0.0) | 1 (0.3) | |
| Vitamin D RCT (%) | | | 0.007 |
| Yes | 134 (39.2) | 150 (43.9) | |
| No | 139 (40.6) | 153 (44.7) | |
| Not Participated | 69 (20.2) | 39 (11.4) | |
| Any Neurodevelopmental Disorder Diagnosis (%) | 32 (10.9) | 54 (18.8) | 0.011 |
| Maternal Intelligence PRS (mean (SD)) | 0.10 (1.01) | −0.09 (0.95) | 0.012 |
| Child Intelligence PRS (mean (SD)) | 0.08 (0.94) | −0.07 (1.10) | 0.068 |
| Head Circumference PRS (mean (SD)) | −0.02 (1.00) | 0.02 (1.00) | 0.565 |

dietary exposures in our observational data (Figs 1C and S2). Detailed correlations can be visualised in a comprehensive heatmap (S3 Fig).

## Dietary patterns in pregnancy are associated with child cognition at 2.5 and 10 years

The mean CCS, assessed at 2.5 years, was 104.8 (sd = 9.7). Scores were higher amongst girls than boys (105.9 (10.3) versus 103.7 (9.1) (*t* test, *p* = 0.004)). FSIQ population mean, assessed at 10 years, was 102.7 (12.1). Girls had higher FSIQ cognitive scores compared to boys (104.5 (11.3) versus 101.1 (12.6), *t* test, *p* < 0.001).

In univariate analysis, the Western dietary pattern metabolite score in pregnancy (per 1 SD change) was negatively associated with CCS (*β* −1.43 [−2.18, −0.67], *p* < 0.001) and FSIQ at 10

years ($\beta$ −2.45 [−3.42, −1.47], $p < 0.001$). In multivariable analysis, these results were consistent for CCS ($\beta$ −1.24 [−2.16,−0.32], $p = 0.008$), whereas FSIQ no longer reached statistical significance ($\beta$ −0.96 [−2.07,0.15], $p = 0.09$) (Tables 2 and S4 for WISC-IV composite scores). Findings were comparable after further adjusting for genetic confounding.

There was no univariate association between the Varied dietary pattern metabolite score during pregnancy and CCS ($\beta$ 0.42 [−0.35,1.20], $p = 0.280$), but a significant association with FSIQ ($\beta$ 2.39 [1.40,3.37], $p < 0.001$). This association remained significant after multivariable adjustment ($\beta$ 1.29 [0.27,2.30], $p = 0.014$), and after further adjusting for genetic confounding. Breastfeeding duration (per SD change) was not univariately associated with CCS ($\beta$ 0.08 [−0.74,0.89], $p = 0.85$), but was associated with FSIQ at age 10 years ($\beta$ 1.47 [0.45,2.50], $p = 0.005$). This finding became insignificant after multivariable adjustment ($\beta$ 0.43 [−0.59,1.44], $p = 0.412$).

Sensitivity analyses adjusting for prenatal nutrient supplementation randomised controlled trials (RCT)s, child Western dietary patterns, parental head circumference, and any neurodevelopmental diagnosis at 10 years did not change the inference of our findings (S5 Table). There were no significant differences found in relation to child sex, as determined through statistical interaction (S6 Table).

In subanalysis, we assessed the temporality of the significant associations between the Western dietary pattern and CCS scores, as well as the Varied dietary pattern and FSIQ scores. To do this, we calculated dietary pattern metabolite scores at four additional postnatal time points: for mothers at 1-week postpartum and for children at 6 months, 18 months (for CCS), and additionally at 6 years (for FSIQ). For the Western dietary pattern, despite collinearity between scores (S7 Table), adding postnatal metabolite scores improved CCS model prediction ($r2$ increase from 0.035 to 0.043, $p = 0.038$). However, FSIQ prediction for the Varied dietary pattern was not improved when considering postnatal metabolite scores ($p = 0.68$).

## Head circumference growth in early life is positively associated with child cognition at 10 years

Head circumference cross-sectionally was not univariately associated with CCS at 2.5 years ($\beta$ −0.33 [−1.14,0.48], $p = 0.420$), but was significantly associated with FSIQ at 10 years ($\beta$ 1.21 [0.25,2.16], $p = 0.013$).

**Table 2. Associations between dietary pattern metabolite scores, duration of breastfeeding and cognition outcomes.** This table presents the results of linear regression analyses assessing the associations between dietary exposures and cognition outcomes; specifically, the Bayleys-III cognitive composite score at 2.5 years and WISC: Full Scale Intelligence Quotient at 10 years. Estimates are interpreted as the effect of a 1 standard deviation increase of a Western, or Varied dietary pattern metabolite score, or logged breastfeeding duration. The table provides both unadjusted and adjusted associations, with the latter controlling for potential confounders including pre-pregnancy maternal body mass index, child sex, birth weight, gestational age, smoking during pregnancy, antibiotic use during pregnancy, pre-eclampsia, household income at birth, maternal education level at birth, and maternal age at birth. Further adjustments include maternal and child polygenic risk score (PRS) for intelligence, and child head circumference PRS.

| Cognitive scores | Univariate model | Multivariable model | Multivariable with PRS |
|---|---|---|---|
| **Western dietary pattern** | **Estimate [95% Cl] p-value** | **Estimate [95% Cl] p-value** | **Estimate [95% Cl] p-value** |
| Bayley-III Composite Score | −1.43 [−2.18, −0.67], $p < 0.001$ | −1.24 [−2.16, −0.32], $p = 0.008$ | −1.24 [−2.16, −0.33], $p = 0.008$ |
| WISC: Full Scale Intelligence Quotient | −2.45 [−3.42, −1.47], $p < 0.001$ | −0.96 [−2.07, 0.15], $p = 0.090$ | −0.98 [−2.05, 0.10], $p = 0.077$ |
| **Varied dietary pattern** | **Estimate [95% Cl] p-value** | **Estimate [95% Cl] p-value** | **Estimate [95% Cl] p-value** |
| Bayley-III Composite Score | 0.42 [−0.35, 1.20], $p = 0.280$ | 0.03 [−0.81, 0.86], $p = 0.953$ | −0.01 [−0.85, 0.83], $p = 0.977$ |
| WISC: Full Scale Intelligence Quotient | 2.39 [1.4, 3.37], $p < 0.001$ | 1.29 [0.27, 2.30], $p = 0.014$ | 1.17 [0.18, 2.17], $p = 0.021$ |
| **Duration of breastfeeding** [*] | **Estimate [95% Cl] p-value** | **Estimate [95% Cl] p-value** | **Estimate [95% Cl] p-value** |
| Bayley-III Composite Score | 0.08 [−0.74, 0.89], $p = 0.854$ | −0.23 [−1.08, 0.62], $p = 0.592$ | −0.19 [−1.04, 0.65], $p = 0.653$ |
| WISC: Full Scale Intelligence Quotient | 1.47 [0.45, 2.50], $p = 0.005$ | 0.43 [−0.59, 1.44], $p = 0.412$ | 0.56 [−0.43, 1.55], $p = 0.271$ |

[*]Note breastfeeding is log-transformed and $z$-scored, thus estimates are interpreted as per 1 SD change.

For the latent class growth trajectory modelling, our analysis focussed on two classes; 'Increasing' (26%) and 'Reference' (74%) (Fig 2). In univariate analysis, children in the 'Increasing' head circumference latent class had 4.87 IQ points higher score at 10 years in FSIQ compared with children in the 'Reference' group ($\beta$ 4.87 [2.62,7.12], $p < 0.001$). This association remained significant in the multivariable model ($\beta$ 3.40 [1.21,5.60], $p = 0.002$) and when further adjusting for genetic confounding (Table 3, see S8 Table for WISC composite scores).

In linear mixed models, the slope, representing head circumference growth (per SD change), was similarly positively associated with FSIQ ($\beta$ 2.03 [1.08,2.97], $p < 0.001$). This association remained significant in multivariable modelling ($\beta$ 1.27 [0.34,2.20], $p = 0.008$) and when further adjusting for genetic confounding (Table 3). Model intercepts were also predictive of FSIQ in univariate ($\beta$ 1.93 [0.94,2.93], $p < 0.001$) and multivariable models ($\beta$ 1.29 [0.33,2.26], $p = 0.009$), and after genetic confounding adjustment (Table 3).

As a sensitivity analysis, we assessed the potential moderation of child genetics scores for head circumference and intelligence, on the association between head circumference and FSIQ at 10 years. In interaction analysis, we found no significant moderating effect of child head circumference PRS ($p$-interaction = 0.442), however, there was a significant and positive moderation of the child intelligence and head circumference on latent classes ($p$-interaction = 0.034), suggesting that for children who have a higher PRS for intelligence, and adhere to the

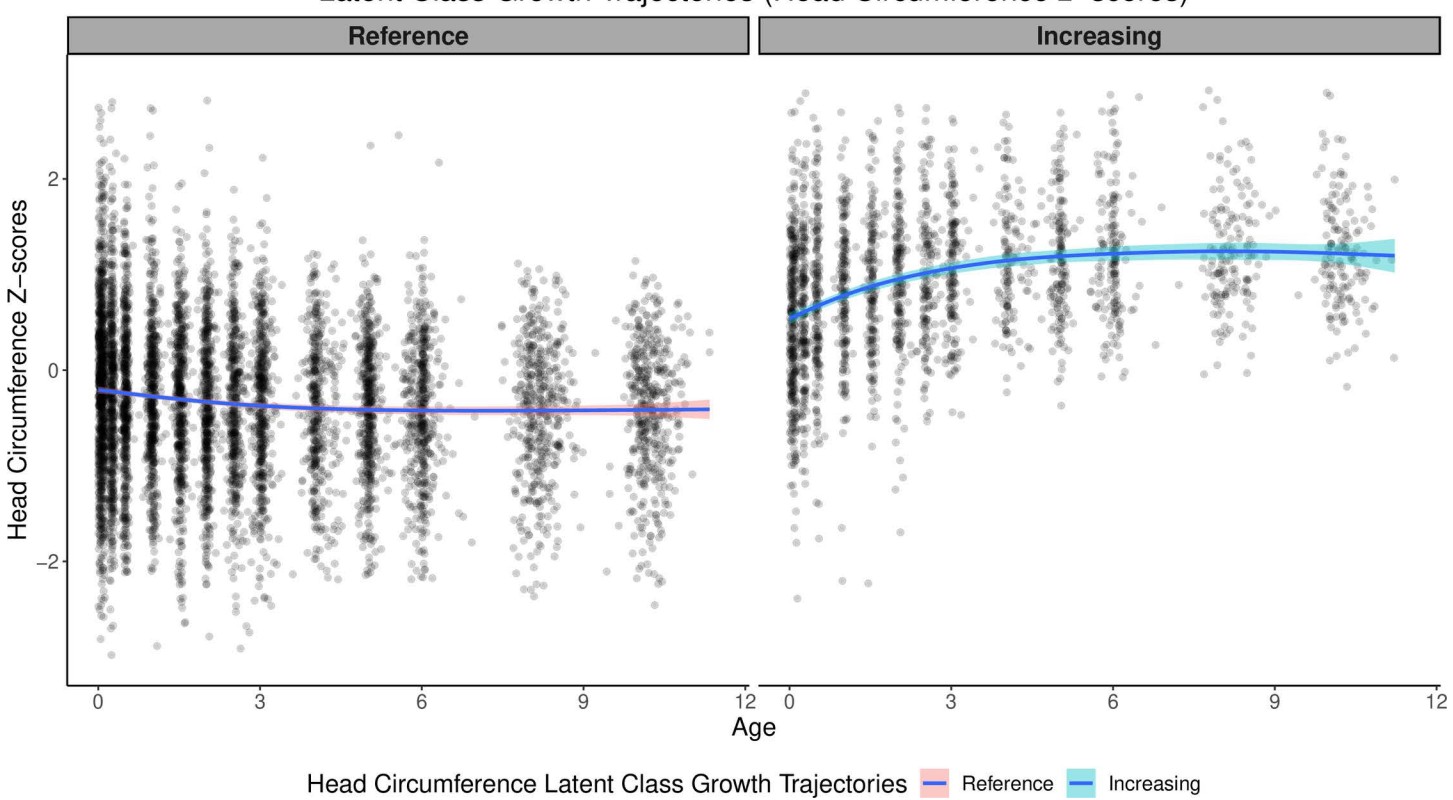

**Fig 2. Latent class trajectories of calibrated head circumference z-scores.** This figure presents the two latent class trajectories analysed in the main analysis, namely "Reference" (74%, $n = 511$) and "Increasing" (26%, $n = 177$). The trajectories show the calibrated head circumference z-scores at 15 time points from the foetal ultrasound scan (20 weeks gestation) to the 10-year clinical visit. Head circumference measurements were adjusted for child length/height, weight, waist size, sex, and gestation-adjusted age at each clinical visit.

Table 3. **Associations between cognitive outcomes and head circumference growth trajectories.** This table presents the results of linear regression analyses assessing the associations between cognitive outcomes and head circumference growth, as determined by latent class trajectory modelling and linear mixed modelling. The latent class trajectory model estimates compare the "Increasing" (26%) class to the "Reference" (74%) class. Estimates for linear mixed model slope and intercept are interpreted as the effect of 1 standard deviation within our population. The table provides both unadjusted and adjusted associations, with the latter controlling for potential confounders such as pre-pregnancy maternal body mass index, child sex, birth weight, gestational age, smoking during pregnancy, antibiotic use during pregnancy, pre-eclampsia, household income at birth, maternal education level at birth and maternal age at birth. Further adjustments include maternal and child polygenic risk score (PRS) for intelligence, and child head circumference PRS.

| | Univariate model | Multivariable model | Multivariable with PRS |
|---|---|---|---|
| **Latent class trajectory model** | **Estimate [95% Cl] *p*-value** | **Estimate [95% Cl] *p*-value** | **Estimate [95% Cl] *p*-value** |
| Bayley-III Composite Score | 0.33 [−1.43, 2.10], $p = 0.712$ | −0.16 [−1.96, 1.64], $p = 0.858$ | −0.36 [−2.17, 1.46], $p = 0.700$ |
| WISC: Full Scale Intelligence Quotient | 4.87 [2.62, 7.12], $p < 0.001$ | 3.40 [1.21, 5.60], $p = 0.002$ | 2.55 [0.38, 4.72], $p = 0.022$ |
| **Linear Mixed Model (Slope)** | **Estimate [95% Cl] *p*-value** | **Estimate [95% Cl] *p*-value** | **Estimate [95% Cl] *p*-value** |
| Bayley-III Composite Score | 0.07 [−0.68, 0.83], $p = 0.851$ | −0.23 [−1.01, 0.54], $p = 0.556$ | −0.29 [−1.08, 0.49], $p = 0.463$ |
| WISC: Full Scale Intelligence Quotient | 2.03 [1.08, 2.97], $p < 0.001$ | 1.27 [0.34, 2.20], $p = 0.008$ | 1.01 [0.09, 1.92], $p = 0.031$ |
| **Linear Mixed Model (Intercept)** | **Estimate [95% Cl] *p*-value** | **Estimate [95% Cl] *p*-value** | **Estimate [95% Cl] *p*-value** |
| Bayley-III Composite Score | 0.05 [−0.71, 0.82], $p = 0.887$ | −0.17 [−0.94, 0.60], $p = 0.668$ | −0.22 [−1.00, 0.55], $p = 0.575$ |
| WISC: Full Scale Intelligence Quotient | 1.93 [0.94, 2.93], $p < 0.001$ | 1.29 [0.33, 2.26], $p = 0.009$ | 0.99 [0.03, 1.94], $p = 0.043$ |

'Increasing' latent class, there is a stronger association with FSIQ (S4A Fig). Moderating effects were stronger in the WISC perceptual reasoning index (*p*-interaction = 0.020) (S4B Fig).

Despite strong associations with FSIQ at age 10 years, we found no association between head circumference growth measures and CCS scores at 2.5 years (Table 3).

## A pregnancy Western dietary pattern is negatively associated and breastfeeding duration positively associated with head circumference growth

In univariate analysis, significant associations were observed between calibrated head circumference at age 10 years and both Western ($\beta$ −0.19 [−0.2,−0.11], $p < 0.0001$) and Varied ($\beta$ 0.17 [0.09,0.25], $p < 0.0001$) dietary patterns during pregnancy, as well as with breastfeeding duration ($\beta$ 0.09 [0.00,0.17], $p = 0.040$). In multivariable longitudinal models, including all 15 time points, we observed that our three dietary variables of interest were all strongly associated with head growth, as indicated by significant interaction terms (*p*-interaction < 0.0001) (S5 Fig). After mutually adjusting for these variables, only the Western dietary pattern (*p*-interaction = 0.020) and breastfeeding duration (*p*-interaction = 0.0001) remained significant (S5 Fig). Therefore, we opted to exclude the Varied dietary pattern from subsequent modelling.

In multivariable linear mixed modelling, both the pregnancy Western dietary pattern (*p*-interaction < 0.0001) and breastfeeding duration (*p*-interaction < 0.0001) independently contributed to head circumference growth over time, with distinct associations (Fig 3 and S9 Table). The pregnancy Western dietary pattern showed a negative association, indicating smaller child head circumference over time, while breastfeeding length exhibited a positive association, suggesting larger head circumference with longer duration of breastfeeding. Comparable results were observed when using uncalibrated head circumference (*p*-interaction < 0.0001) and with head circumference data outliers (1.0%) removed (*p*-interaction < 0.001). Child sex did not significantly moderate the associations of the Western dietary pattern ($p = 0.14$) and breastfeeding length ($p = 0.51$) (S6 Fig).

In subanalysis, adjustments for maternal ($p = 0.003$) and paternal head circumference ($p < 0.0001$) significantly contributed to head circumference growth modelling. Yet, these further adjustments did not alter the estimates between the Western dietary pattern,

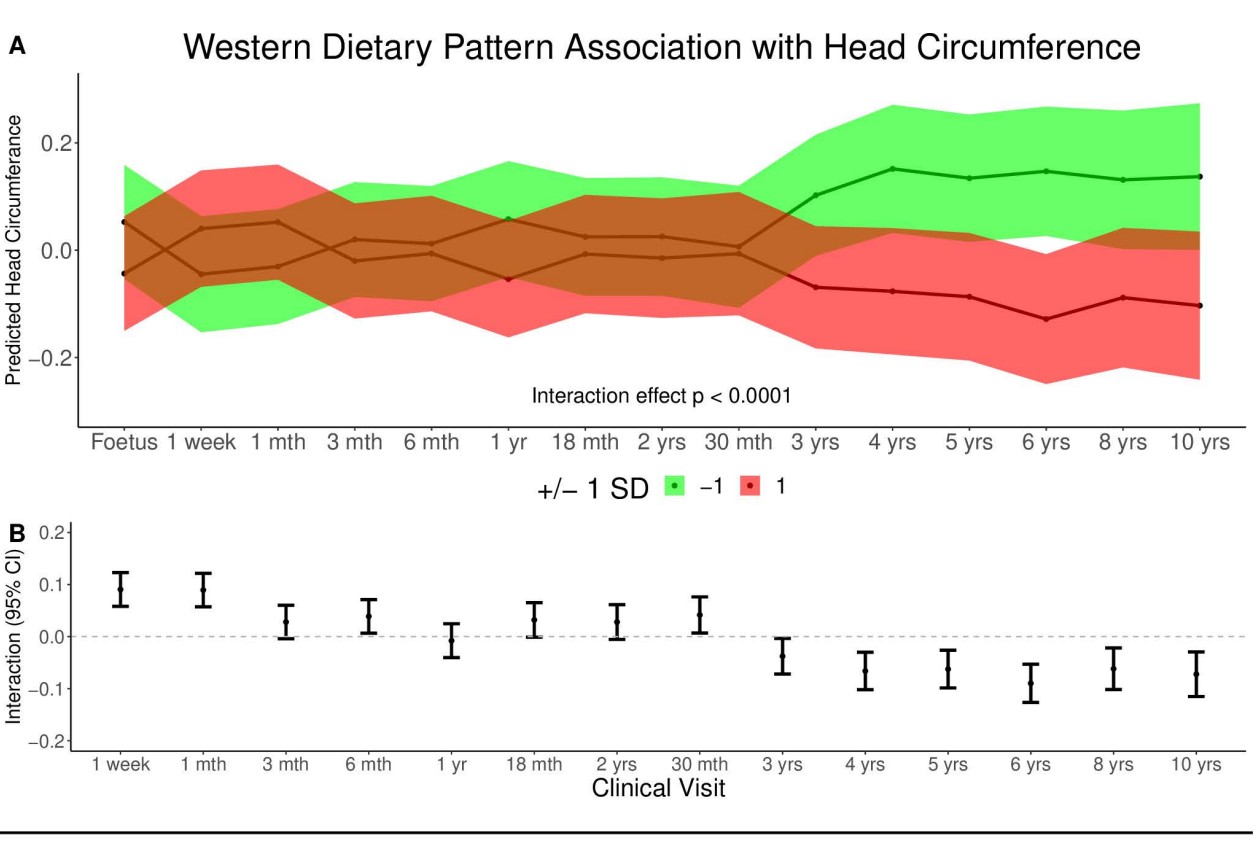

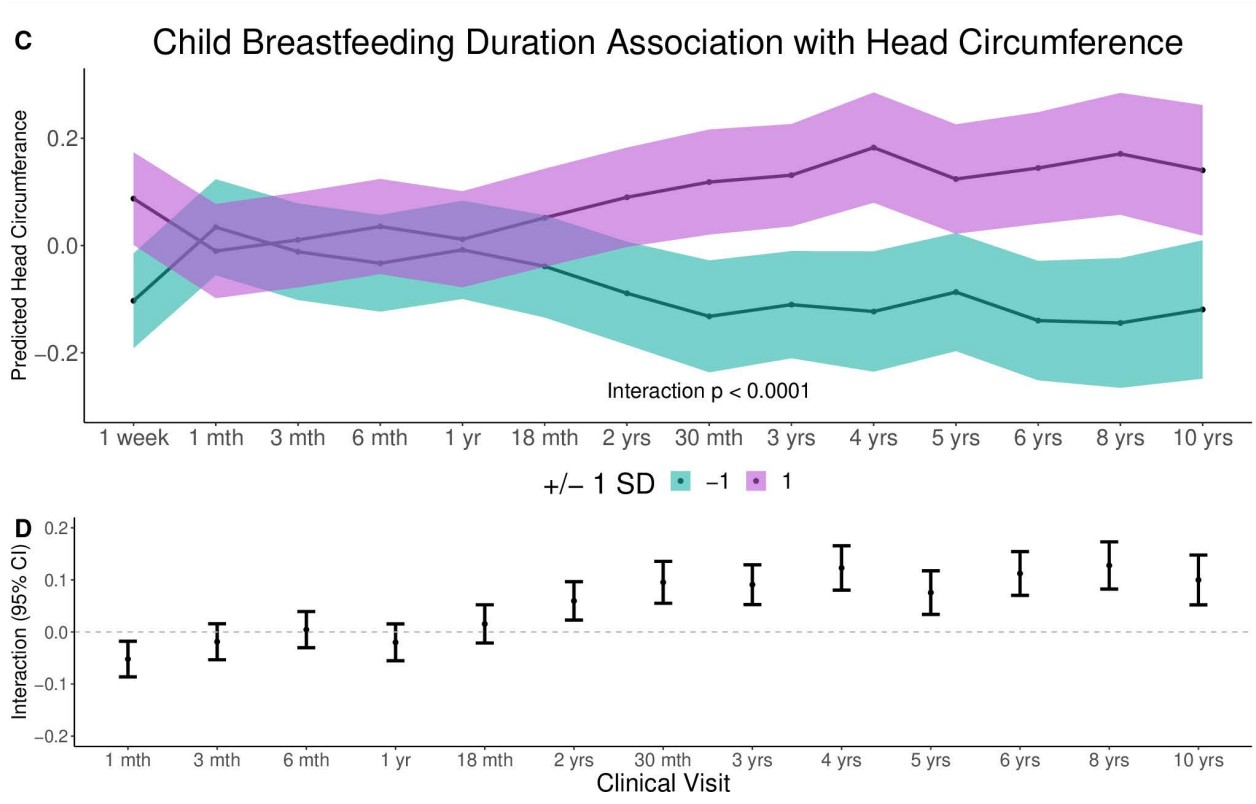

**Fig 3. Predicted head circumference z-scores from Western dietary pattern metabolite score during pregnancy and breastfeeding duration.** This figure presents the predicted head circumference z-scores (with 95% CL) of a Western dietary pattern metabolite score during pregnancy (p < 0.0001) and breastfeeding duration (p < 0.0001) on longitudinal measures of head circumference in a multivariable linear mixed model (± 1 SD

of the exposure). Head circumference measures are adjusted at each visit for child length/height, weight, waist circumference, sex, and age. Panels **A** and **C** show the predicted head circumference estimates with 95% confidence limits for the Western dietary pattern metabolite score during pregnancy and breastfeeding duration, respectively. Panels **B** and **D** show the interaction term effects at each clinical visit for the Western dietary pattern metabolite score during pregnancy and breastfeeding duration, respectively.

breastfeeding duration, and head circumference growth, indicating no confounding from parental traits linked to the inheritable nature of these phenotypes.

To explore the temporal associations of a Western diet on child head growth, we included maternal metabolome-derived Western dietary scores from postnatal stages (1-week postnatal), and children at 6 months, 18 months, and 6 years into our models. Adding these four additional metabolite score time points did not improve our modelling (anova $p = 0.63$). A similar finding was observed for the Varied dietary pattern (anova $p = 0.58$). However postnatal Western dietary pattern metabolite scores showed independent associations with head circumference growth at 1 week postpartum ($p$-interaction $< 0.001$), 6 months ($p$-interaction $= 0.011$), and 18 months ($p$-interaction $= 0.005$), but not at 6 years ($p$-interaction $= 0.904$) (S7 Fig). Varied dietary pattern metabolite scores in children did not show independent associations with head circumference growth ($p$-interaction $> 0.16$) (S7 Fig).

To assess the dynamic genetic contributions to head circumference growth over time, we included child and maternal PRSs for intelligence, and child head circumference as interaction terms with clinical visits, to capture growth trajectories (S8 Fig). Child intelligence PRS (anova $p = 0.014$) and child head circumference PRS (anova $p = 0.006$) independently improved head circumference growth modelling, compared to models without these genetic interaction terms, whereas maternal intelligence PRS (anova $p = 0.51$) showed no improvement. In models mutually adjusted for child intelligence and head circumference PRS, the child intelligence PRS displayed a significant main effect ($p = 0.002$), implying a direct influence on head circumference, while the child head circumference PRS did not reach statistical significance ($p = 0.071$). Interaction effects revealed a differential pattern: the intelligence PRS interaction was not significant ($p$-interaction $= 0.204$), while the head circumference PRS interaction was significant ($p$-interaction $= 0.038$). Notably, the Western dietary pattern and breastfeeding duration interactions persisted ($p$-interaction $\leq 0.0001$), emphasising their independent associations with head circumference, distinct from intelligence or head circumference genetic influences.

### Mediating role of head circumference growth in the association between a varied dietary pattern in pregnancy and child cognition at age 10

In sensitivity analysis, we explored whether head circumference growth may mediate the significant association between the Varied dietary pattern and child cognition at age 10 using causal mediation analysis. We found that head circumference growth significantly mediated this association in multivariable modelling. The Average Causal Mediation Effect (ACME) was 0.182 ($p = 0.022$), accounting for 13.5% of the total effect (Proportion Mediated, $p = 0.034$), indicating a substantial mediation effect.

## Discussion

Our prospective mother–child cohort study identified a nuanced interplay between dietary patterns during pregnancy, head circumference growth trajectories, and cognitive function in children. We identified a strong negative association between a Western dietary pattern in mid-pregnancy, assessed through objective blood metabolomics, with child cognition at 2.5 years of age, and a decrease in growth of the child's head circumference in early life. A Varied

dietary pattern in pregnancy and breastfeeding duration were positively associated with head circumference growth, and our mediation analysis revealed that head circumference growth may serve as a mediator in the positive association between a Varied dietary pattern and estimated IQ at 10 years. Both intelligence and head circumference genetics were independently associated with head circumference growth, yet our dietary findings were independent of these genetic influences, including adjustments for parental head circumference. These findings not only shed light on the intricate links between nutrition and developmental outcomes, but underscore the importance of maternal and early-life dietary choices in relation to cognitive trajectories in children.

Our study resonates with previous research that underlines the significance of diet in promoting neurodevelopment in childhood. The ALSPAC cohort using a clustering methodology to assess pregnancy dietary patterns, found that mothers in the "meat and potatoes" and "white bread and coffee" clusters during pregnancy had offspring with lower IQ at 8 years of age compared to children of mothers in the "fruit and vegetables" cluster [48]. Another study found that adherence to an "aquatic products, fresh vegetables, and homonemeae" pattern in pregnancy was associated with a decreased risk of offspring being 'at risk' or 'emerging' in cognitive and gross motor development at 1 year as assessed by Bayley-III [49]. Our study uses objective dietary assessments in pregnancy to examine associations with postnatal head circumference growth. Previous research has shown positive associations between maternal folate concentrations during pregnancy and foetal head growth [50]. Child nutrition is also known to contribute, where studies have notably found that malnutrition, as measured by infant growth anthropometrics, is negatively associated with child head circumference and cognition [51,52]. We complement these studies by finding predictive value of child dietary metabolite scores and breastfeeding duration with head circumference [51,52]. Finally, our study complements existing research where head circumference growth in early life is an important determinant for later childhood cognition [13,53].

We did not observe a direct association between head growth and cognitive scores at 2.5 years in our study, suggesting that prenatal dietary effects on child cognition in early life may operate through mechanisms not directly linked to changes in brain volume, as indicated by head circumference. Regarding potential mechanisms, literature suggests that maternal gut [54] and child meconium [55] microbiome may be associated with child head circumference growth in early life, and maternal high-fat diets have been shown to have a significant reduction in both maternal and child microbiome diversity [56]. Pre-clinical studies suggest that maternal dysbiosis modulates foetal neurodevelopment [57], further investigations are warranted to understand the interplay between the microbiome and maternal diet on child neurodevelopment.

Our study had many strengths, including the use of pregnancy and early-life samples, enabling us to identify potential risk factors and determine the directionality of associations. We objectively derived dietary patterns from blood metabolomics in pregnancy, based on the results of a FFQ analysed using unbiased PCA, resulting in findings that more accurately reflect the dietary patterns existing in the population [58]. Further, the use of an objectively measured metabolome signature overcomes some of the limitations associated with using FFQs. The increasing utilisation and validation of metabolomics for assessing dietary exposures underscores the growing importance of this method in enhancing the reliability and accuracy of dietary assessments [59]. Accordingly, we have in previous work externally validated the Western dietary pattern metabolite score in an independent cohort [29]. Leveraging the longitudinal design of our study with repeated measures in our metabolome datasets, alongside the comprehensive data and detailed phenotyping from the COPSAC2010 cohort, we were able to facilitate temporal analysis and genetic characterisation. This enabled

us to demonstrate that our findings were independent of genetic contributions. Moreover, adjustments for parental head circumference did not alter our findings, reinforcing that the observed dietary effects on child growth are not confounded by highly heritable genetic factors related to parental traits [21,60]. Regarding the limitations of our observational study, despite extensive adjustments for genetic and phenotypic factors, residual confounding due to unmeasured variables related to maternal or child phenotypes cannot be entirely excluded, and some associations may be spurious; however, supplemental analysis adjusting for maternal genetic risk for intelligence did not change our findings. Our observational data do not allow us to elucidate the critical window, if any, of these dietary effects. Nevertheless, the strong coefficients between maternal and child Western dietary metabolome signatures give us confidence in our modelling, and previous research suggests that diets prior to and during pregnancy tend to be stable [61]. Finally, our findings should only be interpreted in the context of a developed country where the study was set. Previous studies in mostly low- to middle- income countries have identified malnutrition and stunting as key determinants of head circumference growth and cognition outcomes in children [62–64]. This underscores the imperative role of ensuring proper nutritional intake during early childhood. Finally, in the context of modelling head circumference growth, child metabolome Western dietary pattern scores independently predicted growth up to 18 months, indicating their potential as an objective clinical tool for monitoring growth.

Our research emphasises the importance of healthy dietary patterns during pregnancy and breastfeeding for optimal offspring neurodevelopment. We report a negative association between a Western dietary pattern during pregnancy and offspring cognition and head circumference growth in early life. Our findings support the hypothesis of a prenatal programming effect of a Western dietary pattern during pregnancy on offspring neurodevelopment. Further research is needed to elucidate the mechanisms underlying these observed associations. Overall, the findings of this study have meaningful public health implications, emphasising the need for intensifying efforts to encourage healthy and balanced dietary patterns during pregnancy, and promoting breastfeeding practices, to optimise offspring neurodevelopment.

## Governance

We are aware of and comply with recognised codes of good research practice, including the Danish Code of Conduct for Research Integrity. We comply with national and international rules on the safety and rights of patients and healthy subjects, including Good Clinical Practice as defined in the EU's Directive on Good Clinical Practice, the International Conference on Harmonisation's good clinical practice guidelines and the Helsinki Declaration. We follow national and international legislation on General Data Protection Regulation (GDPR), the Danish Act on Processing of Personal Data and the practice of the Danish Data Inspectorate.

## Supporting information

**S1 Table. Nutrient constituents and principal component loadings for dietary patterns.** (DOCX)

**S2 Table. Baseline characteristics and cognition scores stratified by participation in COPSYCH 10-year clinical visit.** (DOCX)

**S3 Table. Baseline characteristics and cognition scores stratified by child sex.** (DOCX)

**S4 Table. Associations between Western dietary pattern metabolite score and WISC-IV composite scores.**
(DOCX)

**S5 Table. Subanalysis; multivariable regression modelling of dietary exposures and cognitive outcomes.**
(DOCX)

**S6 Table. Sex-stratified linear regression modelling of dietary exposures and cognitive outcomes.**
(DOCX)

**S7 Table. Correlation coefficients between the Western dietary pattern scores for maternal blood metabolomes and child blood metabolomes.**
(DOCX)

**S8 Table. Western dietary pattern metabolite score during pregnancy and head circumference growth: latent class and linear mixed model results.**
(DOCX)

**S9 Table. ANOVA table for the multivariable linear mixed model of Western dietary pattern metabolite score during pregnancy and breastfeeding duration.**
(DOCX)

**S1 Fig. Principal component loadings for the Western and varied dietary patterns.**
(DOCX)

**S2 Fig. Three-dimensional scatterplot illustrating the associations between maternal diet, cognition, and head circumference growth.**
(DOCX)

**S3 Fig. Heatmap of correlations between dietary exposures, outcomes, and model covariates.**
(DOCX)

**S4 Fig. modulating effects of genetics scores for head circumference and intelligence on the association between head circumference and cognition at 10 years.**
(DOCX)

**S5 Fig. Independent and mutual interaction term effects of dietary exposures.**
(DOCX)

**S6 Fig. Sex Stratified effects of a pregnancy Western dietary pattern and breastfeeding duration on longitudinal measures of head circumference in a multivariable linear mixed model.**
(DOCX)

**S7 Fig. Independent effects of the Western and Varied dietary metabolite scores for mothers 1-week postpartum and children's metabolome at various time points on head circumference.**
(DOCX)

**S8 Fig. Effect of child intelligence and head circumference polygenic risk score on longitudinal measures of head circumference.**
(DOCX)

**S1 STROBE Checklist.  Checklist outlining the items that should be included in reports of cohort studies, following the STROBE reporting guidelines.**
(DOCX)

**S1 File.  Supplementary methods.**
(PDF)

## Author contributions

**Conceptualisation:** David Horner, Jens Richardt Møllegaard Jepsen, Bo Chawes, Louise Monnerup, Klaus Bønnelykke, Bjørn H Ebdrup, Jakob Stokholm, Morten Arendt Rasmussen.

**Data curation:** David Horner, Jens Richardt Møllegaard Jepsen, Bo Chawes, Julie B Rosenberg, Parisa Mohammadzadeh, Yang Luo, Janine F Felix, Klaus Bønnelykke, Bjørn H Ebdrup, Jakob Stokholm, Morten Arendt Rasmussen.

**Formal analysis:** David Horner, Yang Luo, Bjørn H Ebdrup, Jakob Stokholm, Morten Arendt Rasmussen.

**Funding acquisition:** David Horner, Bo Chawes, Birgitte Fagerlund, Birte Y Glenthøj, Klaus Bønnelykke, Bjørn H Ebdrup, Jakob Stokholm, Morten Arendt Rasmussen.

**Investigation:** David Horner, Bjørn H Ebdrup, Jakob Stokholm, Morten Arendt Rasmussen.

**Methodology:** David Horner, Klaus Bønnelykke, Bjørn H Ebdrup, Jakob Stokholm, Morten Arendt Rasmussen.

**Project administration:** David Horner, Bjørn H Ebdrup, Jakob Stokholm, Morten Arendt Rasmussen.

**Resources:** David Horner, Morten Arendt Rasmussen.

**Software:** David Horner, Morten Arendt Rasmussen.

**Supervision:** David Horner, Bjørn H Ebdrup, Jakob Stokholm, Morten Arendt Rasmussen.

**Validation:** David Horner, Jakob Stokholm, Morten Arendt Rasmussen.

**Visualisation:** David Horner, Bjørn H Ebdrup, Jakob Stokholm, Morten Arendt Rasmussen.

**Writing – original draft:** David Horner, Jens Richardt Møllegaard Jepsen, Bo Chawes, Klaus Bønnelykke, Bjørn H Ebdrup, Jakob Stokholm, Morten Arendt Rasmussen.

**Writing – review & editing:** David Horner, Jens Richardt Møllegaard Jepsen, Bo Chawes, Rebecca Vinding, Julie B Rosenberg, Parisa Mohammadzadeh, Yang Luo, Birgitte Fagerlund, Trine Flensborg-Madsen, Thomas Ragnar Wood, Janine F Felix, Louise Monnerup, Birte Y Glenthøj, Klaus Bønnelykke, Bjørn H Ebdrup, Jakob Stokholm, Morten Arendt Rasmussen.

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
