## [Editor Report · Decision Letter 0]

31 Jul 2024

Dear Dr Horner, 

Thank you for submitting your manuscript entitled "Dietary Patterns in Pregnancy and Child Breastfeeding Duration are Associated with Cognitive Function and Head Circumference Growth in Childhood" for consideration by PLOS Medicine.

Your manuscript has now been evaluated by the PLOS Medicine editorial staff and I am writing to let you know that we would like to send your submission out for external peer review.

Please re-submit your manuscript within two working days, i.e. by Aug 02 2024 11:59PM. Please do let me know if you need more time (ssunny@plos.org).

Kind regards,

Syba

Syba Sunny, MBBS, MRes, FRCPath

Associate Editor

PLOS Medicine

ssunny@plos.org

---

## [Decision Letter · Decision Letter 1]

11 Nov 2024

Dear Dr Horner,

Many thanks for submitting your manuscript "Dietary Patterns in Pregnancy and Child Breastfeeding Duration are Associated with Cognitive Function and Head Circumference Growth in Childhood" (PMEDICINE-D-24-02450R1) to PLOS Medicine. The paper has been reviewed by subject experts and a statistician; their comments are included below and can also be accessed here: [LINK]

As you will see, the reviewers were generally positive about the paper, but also listed several suggestions for improvement. After discussing the paper with the editorial team and an academic editor with relevant expertise, I'm pleased to invite you to revise the paper in response to the reviewers' comments. We plan to send the revised paper to some or all of the original reviewers, and we cannot provide any guarantees at this stage regarding publication.

We ask that you submit your revision by Dec 02 2024 11:59PM. However, if this deadline is not feasible, please contact me by email, and we can discuss a suitable alternative.

Don't hesitate to contact me directly with any questions (ssunny@plos.org). 

Best regards, 

Syba 

Syba Sunny, MBBS, MRes, FRCPath 

Associate Editor

PLOS Medicine

ssunny@plos.org

Comments from the academic editor:

The academic editor supported the decision to move forward to a major revision outcome. He commented that he thought your manuscript was ‘overall well written and balanced’. However, he asked that the authors add a bit more to their main text about residual confounding and risk of spurious associations.

Comments from the reviewers: 

Reviewer #1: "Dietary Patterns in Pregnancy and Child Breastfeeding Duration are Associated with Cognitive Function and Head Circumference Growth in Childhood" studies how maternal dietary patterns (defined as "varied"/"Western") during pregnancy and duration of breastfeeding, are associated with child head circumference and cognitive outcomes, on 700 mother-child pairs from Denmark. It was concluded that a Western dietary pattern is associated with lower cognitive scores/decreased head circumference growth, as opposed to a varied dietary pattern.

Some issues might be considered:

1. In Line 163, it is stated that the first principal component reflected a varied dietary pattern (44.3% of participants), and the second component reflected a Western dietary pattern (10.7%). Details about how the principal components were analysed/obtained, and how each participant was eventually assigned to varied/Western dietary pattern (from the metabolite scores & questionnaire, and the meaning of the dietary pattern axes in Supplementary Figure S1 etc.), should be provided given its centrality to the study. While Figure 1 shows some information, it remains unclear as to how a participant would be assigned to varied/Western.

2. Moreover, it might be discussed as to how only varied/Western dietary patterns were defined - other dietary patterns (e.g. vegan, vegetarian, etc.) would seem plausible too.

3. From the covariates presented in the section beginning Line 214, possibly relevant variables such as ethnicity and parental (including paternal) intelligence/IQ (this perhaps estimated with maternal education as a proxy) appear to be missing. Authors might discuss the most relevant variables affecting child head circumference/cognitive outcomes from prior work, since it is currently only briefly acknowledged that heritability estimates for intelligence are high (if lower in childhood), but without context of their relative impact.

4. For Supplementary Figure S2, it might be considered to briefly explain the linear mixed model parameter & latent class variables - which outcome were they for?

5. Related to the above, it would be interesting to consider the raw correlations between the two dietary patterns and the other variables, if possible.

6. While it is stated that 700 mother-child pairs were involved, Table 1 (n=684) and Supplementary Tables S1 & S2 (n=695) give different cohort sizes. This might be clarified, possibly alongisde the inclusion of a flowchart detailing any exclusion steps.

Reviewer #2: This study examined the associations between maternal dietary pattern during pregnancy and breastfeeding and children's cognitive development. Their results suggest that the Western dietary pattern during pregnancy was associated with lower cognitive scores at 2.5 y and less head circumference growth while a varied dietary pattern was associated with higher IQ at 10y and head circumference growth as a contributor. The study was well-conducted and the longitudinal nature of the study provided valuable novel evidence regarding the long-lasting influence of maternal diet on cognitive outcomes of offspring. Here are some questions: 

Line 157-168: It was confusing how the metabolomics and FFQ data were combined together to generate a dietary pattern. What was the evidence base that those metabolites were reflective of dietary exposure and how it was reconciled with the food frequency from FFQ? 

Line 215-221: Was any postnatal exposure controlled for as covariates such as socioeconomic status of the family during childhood, children's medical history, early education program attendance, etc. 

The study also mentioned attendance of two RCTs. What was the proportion of participation who participated in RCTs? Can the excess exposure to nutrients included in the RCTs be adjusted for? 

Reviewer #3: Thank you for the opportunity for reviewing this interesting manuscript. The study is well designed, and the paper is well-written. I have a few comments for the introduction and methods sections for the authors to improve the clarification of this manuscript. 

Introduction: 

* The authors mentioned "… but previous research has not adequately disentangled the potential independent, additive or pleiotropic effects of maternal and child dietary intake on brain morphological or cognitive outcomes." It would be helpful to list out what's known from the literature in a little bit more detail, so the readers could understand the current research gap. 

* There is no mention of genetics in the intro. You may consider provide a brief summary of the literature on the genetic influence. 

Methods:

* Is the metabolite score validated? If so, against what? It would be helpful to provide some background as the metabolite score is not widely used or known.

* The dietary exposure section may require a bit mor details/ clarification. How was the metabolite score integrated with the dietary data collected in the FFQ? The authors mentioned the scores were modeled on dietary patterns identified by the FFQ. Was the residuals used for analysis? Later in the paragraph, the authors included "the metabolite scores were internally z-scored", which seems to serve as an independent exposure without integrating with the FFQ data. In general, the authors are expected to clarify how the metabolite information, FFQ data, were used to define the dietary patterns, as It was a bit unclear to me.

* The polygenic risk score section also need some more details. What loci were used to calculate the PRS? Was the method previously validated in your study population? How relevant was the PRS to your outcome of interest? Some of the information can be provided in the introduction, echoing a previous comment. 

* Covariates: why not considering parity as a covariate? It would be also important to clarify if the maternal dietary pattern was controlled for when assessing the relationships between child dietary score/ duration of breastfeeding and the outcomes, and vice versa.

---

* Please upload any figures associated with your paper as individual TIF or EPS files with 300dpi resolution at resubmission; please read our figure guidelines for more information on our requirements: http://journals.plos.org/plosmedicine/s/figures. While revising your submission, please upload your figure files to the PACE digital diagnostic tool, https://pacev2.apexcovantage.com/. PACE helps ensure that figures meet PLOS requirements. To use PACE, you must first register as a user. Then, login and navigate to the UPLOAD tab, where you will find detailed instructions on how to use the tool. If you encounter any issues or have any questions when using PACE, please email us at PLOSMedicine@plos.org.

* Thank you for providing a data availability statement. For data residing with a third party, authors are required to provide instructions with contact information (web or email address) for obtaining the data. Please note that a study author cannot be the contact person for the data. 

* Please include page numbers (as well as line numbers) in the manuscript file. 

FIGURES AND TABLES

SUPPLEMENTARY MATERIAL

* Please note that supplementary material will be posted as supplied by the authors. 

REFERENCES

OBSERVATIONAL STUDIES

* Abstract: Please include the study design, population and setting, number of participants, years during which the study took place (enrollment and follow up), length of follow up, and main outcome measures.

* Please ensure that the study is reported according to the STROBE (or appropriate STOBE extension) guideline (available from: https://www.equator-network.org/reporting-guidelines/strobe) and include the completed STROBE (or STROBE extension) checklist as Supporting Information. Please add the following statement, or similar, to the Methods: "This study is reported as per the Strengthening the Reporting of Observational Studies in Epidemiology (STROBE) guideline (S1 Checklist)." When completing the checklist, please use section and paragraph numbers, rather than page numbers. 

* For all observational studies, in the manuscript text, please indicate: (1) the specific hypotheses you intended to test, (2) the analytical methods by which you planned to test them, (3) the analyses you actually performed, and (4) when reported analyses differ from those that were planned, transparent explanations for differences that affect the reliability of the study's results. If a reported analysis was performed based on an interesting but unanticipated pattern in the data, please be clear that the analysis was data driven. 

* Please state in the Methods section whether the study had a prospective protocol or analysis plan. If a prospective analysis plan (from your funding proposal, IRB or other ethics committee submission, study protocol, or other planning document written before analyzing the data) was used in designing the study, please include the relevant document(s) with your revised manuscript as a Supporting Information file to be published alongside your study and cite it in the Methods section. A legend for this file should be included at the end of your manuscript. If no such document exists, please make sure that the Methods section transparently describes when analyses were planned, and when/why any data-driven changes to analyses took place. Changes in the analysis, including those made in response to peer review comments, should be identified as such in the Methods section of the paper, with rationale.

---

## [Decision Letter · Decision Letter 2]

28 Jan 2025

Dear Dr. Horner,

Thank you very much for re-submitting your manuscript "Dietary Patterns in Pregnancy and Child Breastfeeding Duration are Associated with Cognitive Function and Head Circumference Growth in Childhood" (PMEDICINE-D-24-02450R2) for review by PLOS Medicine.

I have discussed the paper with my colleagues and the academic editor and it was also seen again by 2 of the original reviewers (including the statistician). The statistical reviewer has some further suggestions for improvement. However, I am pleased to say that, provided the remaining comments from the statistician are addressed, and the editorial and production issues are dealt with, we are planning to accept the paper for publication in the journal. Please note that we plan to send your next revision to the statistical reviewer for further comments.

The remaining issues that need to be addressed are listed at the end of this email. 

We expect to receive your revised manuscript within 1 week. Please email us (plosmedicine@plos.org) if you need more time, or have any questions or concerns.

We look forward to receiving the revised manuscript by Feb 04 2025 11:59PM. 

Sincerely,

Syba

Syba Sunny, MBBS, MRes, FRCPath

Associate Editor 

PLOS Medicine

plosmedicine.org

Requests from Editors:

We are very pleased to be moving forward with your manuscript. As previously mentioned, we ask that you address the statistical reviewer’s comments (Reviewer 1 – see comment further down) and provide point-by-point responses. In addition to this, we have one editorial request: we ask that you amend your STROBE checklist slightly - please delete the page numbers in the right hand column and replace these with section and paragraph numbers.

Comments from Reviewers:

Reviewer #1: We thank the authors for addressing our previous comments. The major remaining clarification would be on the choice of PCA for analysis of dietary patterns, as inquired in some form by all reviewers.

Since PCA is an unsupervised learning method, to the best of our knowledge, it is not automatically guaranteed that the first and second (or any particular) principal components derived by PCA, would have any particular relation to any factor present in the data. In other words, what was done was to first apply PCA to the energy/macronutrient/micronutrient data obtained from individual questionnaires. After that, the relation to broad dietary patterns was then derived according to the top principal components.

While some clarifications have been made on the PCA analysis, additional details may be warranted given the centrality of the dietary patterns obtained from the analysis:

1. In Line 194, it is stated that dietary patterns were identified based on "calculated estimates of energy, macronutrients, and micronutrients". Given this, a justification for how 'Varied' and 'Western' dietary patterns were originally defined, would be illuminating. For example, what would be the typical/average values for each feature (energy and nutrients), for each of these dietary patterns? The supplementary material refers to using the "FFQ-derived pregnancy Western and Varied dietary patterns as the response variable", but this does not address how the dietary patterns were defined in the first place.

2. In Authors Response #2, it is stated that "an individual with a high scoring for PC1 (e.g. 2), would be considered as having a more 'Varied dietary pattern', and individuals with a lower scoring for PC1 (e.g. -2) would be considered as having a low 'Varied dietary pattern'". It is supposed that the same applies for PC2 and the Western dietary pattern (Line 196). If so, it might be clarified as to the case where an individual subject has relatively high (or low) values for both PC1 and PC2 - again, how would that subject be assigned to 'Varied' or 'Western'?

Reviewer #2: The authors have addressed my questions. Thank you!

[LINK]

---

## [Decision Letter · Decision Letter 3]

12 Feb 2025

Dear Dr. Horner,

Thank you very much for re-submitting your manuscript "Dietary Patterns in Pregnancy and Child Breastfeeding Duration are Associated with Cognitive Function and Head Circumference Growth in Childhood" (PMEDICINE-D-24-02450R3) for review by PLOS Medicine.

I have discussed the paper with my colleagues and it was also seen again by the statistical reviewer. Following a close read of the manuscript, we have identified some additional editorial requests, but I am pleased to say that provided the remaining editorial and production issues are dealt with we are planning to accept the paper for publication in the journal.

The remaining issues that need to be addressed are listed at the end of this email. 

In revising the manuscript for further consideration here, please ensure you address the specific points made by the editors. In your rebuttal letter you should indicate your response to the editors' comments and the changes you have made in the manuscript. Please submit a clean version of the paper as the main article file. A version with changes marked must also be uploaded as a marked up manuscript file.

We look forward to receiving the revised manuscript by Feb 19 2025 11:59PM.   

Sincerely,

Rebecca Kirk

Senior Editor 

PLOS Medicine

plosmedicine.org

Requests from Editors:

GENERAL EDITORIAL REQUESTS

* Please update so that your title complies with to PLOS Medicine's style. Your title must be nondeclarative and not a question. It should begin with main concept if possible. "Effect of" should be used only if causality can be inferred, i.e., for an RCT. Please place the study design ("A randomized controlled trial," "A retrospective study," "A modelling study," etc.) in the subtitle (ie, after a colon).

* Please confirm that your abstract complies with our requirements, including providing all the information relevant to this study type https://journals.plos.org/plosmedicine/s/submission-guidelines#loc-abstract

* Please ensure that all abbreviations are defined at first use throughout the text.

GENERAL

* Please review your text for claims of novelty or primacy (e.g. 'for the first time') and remove this language. In addition, please check that any use of statistical terms (such as trend, effect or significant) are supported by the data, and if not please remove them (this includes in the figures).

* Please separate your confidence intervals with a comma, using a dash can be confused with the minus sign.

* Please remove the 'conclusions' subheading under discussion.

* Information provided in the Supplementary Information will be published 'as is', please check this thoroughly and do not include information that is not for publication in this document.

ETHICS AND CONSENT

* Please specify whether informed consent in this cohort was written or oral. Please ensure that the research complies with the PLOS policy in full: https://journals.plos.org/plosmedicine/s/human-subjects-research#loc-patient-privacy-and-informed-consent-for-publication

FIGURES

* Figures cannot be reproduced from other sources that are not CC-BY and so the additional information that was provided cannot be published in the article. Please provide a different depiction or address the information using an alternative method. This includes for Figure S1.

* Please consider avoiding the use of red and green in order to make your figure more accessible

OBSERVATIONAL, COHORT, CROSS-SECTIONAL, AND CASE CONTROL STUDIES

* Did your study have a prospective protocol or analysis plan? Please state this (either way) early in the Methods section.

c) In either case, changes in the analysis-- including those made in response to peer review comments-- should be identified as such in the Methods section of the paper, with rationale."

* Your study is observational and therefore causality cannot be inferred. Please remove language that implies causality (e.g. effect is used in the figure) and refer to associations instead.

Comments from Reviewers:

Reviewer #1: We thank the authors for addressing our remaining concerns, and for the detailed explanation of the principal component associations. This should be extremely helpful for readers in interpreting the content.

[LINK]

---

## [Editor Report · Decision Letter 4]

4 Mar 2025

Dear Dr Horner, 

On behalf of my colleagues and the Academic Editor, Lars Åke Persson, I am pleased to inform you that we have agreed to publish your manuscript "Maternal Dietary Patterns, Breastfeeding Duration, and Their Association with Child Cognitive Function and Head Circumference Growth: A Prospective Mother-Child Cohort Study" (PMEDICINE-D-24-02450R4) in PLOS Medicine.

Before your manuscript can be formally accepted you will need to complete some formatting changes, which you will receive in a follow up email. Please be aware that it may take several days for you to receive this email; during this time no action is required by you. Once you have received these formatting requests, please note that your manuscript will not be scheduled for publication until you have made the required changes. You will also have the opportunity to update your preprint to the cite the published version in the manuscript at that time.

PRESS

Sincerely, 

Rebecca Kirk 

Senior Editor 

PLOS Medicine